# The Preseason Warming of the Indian Ocean Resulting in Soybean Failure in U.S.

Menghan Li [1,3], Xichen Li [1,2], Yi Zhou [1,3], and Yurong Hou [1,3]

[1]International Center for Climate and Environment Sciences, Institute of Atmospheric Physics, Chinese Academy of Sciences, Beijing, 100193, China.
[2]Institute of Ocean Research, Peking University, Beijing, 100193, China.
[3]University of Chinese Academy of Sciences, Beijing, 100193, China.

*Correspondence to*: Xichen Li (xichenli@pku.edu.cn)

**Abstract.** Soybean is the most important oilseed and feed crop globally. As one of the major soybean producers in the world, soybean yield variability in the United States has garnered widespread attention. We analyze the effect of the Indian Ocean sea surface temperature (SST) on soybean yield variability. Our findings indicate that variations in Indian Ocean SST during the November-December-January (hereinafter referred to as ND(-1)J period, approximately nine months prior to harvest, account for 16% of the anomaly in U.S. soybean yields. Furthermore, for each standard deviation change in the Indian Ocean Basin (IOB) index, there is an estimated 4.0% change in total soybean production in the United States. The root zone soil moisture and maximum temperature during the reproductive growth stage in summer are the key factors influencing the United States soybean yields. The warming of the Indian Ocean could cause hot and dry conditions during July-August-September (JAS) by influencing ND(-1)J soil moisture and the eastern Pacific SST, leading to substantial soybean failures in the United States. Our findings emphasize the importance of the Indian Ocean SST on soybean production in the United States and reveal the pathways of this impact, which can help predict the United States soybean failures and improve food security worldwide.

## 1 Introduction

Soybeans are a vital oilseed and feed crop, providing essential proteins and fats for both human and animal nutrition worldwide. Due to their significant role, the distribution and trade of soybeans have attracted considerable attention, especially given the concentration of production in a few key countries. According to the Food and Agriculture Organization of the United Nations, in 2017, the United States, Brazil, and Argentina together produced approximately 75.8% of the global soybean supply, with the United States being the largest contributor. These three countries also dominate global soybean exports, accounting for over 80% of the total, with the U.S. alone representing more than 40% of exports between 2008 and 2012 (Birgit Meade et al., 2016; Torreggiani et al., 2018). This concentration underscores the importance of production fluctuations in these major soybean-producing nations.

Crop yields are influenced by various local meteorological factors, such as temperature, precipitation, solar radiation, soil moisture, and air humidity (Gaupp et al., 2020; Joshi et al., 2021; Li et al., 2019; Ray et al., 2015). However, for soybeans, hot and dry conditions during the critical reproductive phase in summer (July-August-September, JAS) are key factors leading to reduced yields (Hamed et al., 2021; Nendel et al., 2023; Teasdale and Cavigelli, 2017). For instance, Schauberger et al. (2017)

have investigated the impact of high temperatures on the yields of US soybeans, finding that each day with temperatures above 30°C reduces soybean yields by up to 6% under rainfed conditions and conclude that high temperatures negatively affect US crop yields primarily through water stress mechanisms. The risk of yield loss for soybeans due to drought is especially significant in the USA, with the likelihood of such losses potentially exceeding 70% during an exceptional drought (Leng and Hall, 2019). For example, the late summer of 2012 drought across the central U.S. caused soybean yields across the western Corn Belt and central Plains to be at least 25 percent below the long-term trend (Otkin et al., 2016).

Beyond immediate weather patterns, large-scale climate systems like the El Niño-Southern Oscillation (ENSO) also influence soybean yields by modulating seasonal temperature and precipitation patterns. Persistent La Niña phases, for example, can lead to soybean harvest failures across the U.S. and southeastern South America due to impacts on extratropical sea surface temperatures (SST) and spring soil moisture levels (Hamed et al., 2023). The effect of ENSO on soybean production has been

widely studied (Anderson et al., 2017b; Cao et al., 2023; Nóia Júnior et al., 2020; Perondi et al., 2022), revealing its significant role in soybean production.

Other climate oscillations, including the Atlantic Multidecadal Oscillation (AMO), Indian Ocean Basin (IOB), and North Atlantic Oscillation (NAO), also appear to influence weather conditions that affect soybean yields in the Americas (Gan et al., 2019; Manthos et al., 2022; Wang et al., 2013; Zhao and Brissette, 2022). For instance, a positive AMO phase is associated

with warmer temperatures across North America, especially in the eastern regions (Zhao and Brissette, 2022). Meanwhile, warming of the tropical Indian Ocean basin corresponds with higher winter precipitation over southeastern South America, indicating the IOB's significant influence on regional weather patterns (Hu et al., 2023). Despite these insights, the specific effects of these climate modes on soybean yields in the U.S. remain underexplored.

In this study, we investigated the correlation between multiple climate variability factors and U.S. soybean yields, identifying

a significant influence from Indian Ocean SST anomalies. To quantify the direct and indirect impacts of IOB warming on U.S. soybean production, we first pinpointed the key climatic factors affecting yields. We then linked the Indian Ocean SST with these key factors and analyzed their teleconnection to elucidate its relationship with soybean yields in the United States. Our findings provide a foundation for predicting soybean crop risks and for developing strategies to strengthen resilience in global soybean trade, thereby helping mitigate climate-related risks to food security.

## 2 Materials and Methods

### 2.1 Materials

The U.S. crop data used in this study includes harvest area, production, yield time series, and crop calendars, covering a 42-year period from 1978 to 2019. State-level soybean harvest data was obtained from the USDA's NASS Quick Stats database (https://quickstats.nass.usda.gov/), while planting and harvest dates at a 0.5-minute grid resolution were sourced from Sacks et al. (Sacks et al., 2010). Across the U.S., the primary soybean growing season typically spans May to November.

The annual U.S. soybean yield ($Y_t$) was calculated by aggregating yields across U.S. states, weighted by harvested area. The formula is as follows:

$$Y_t = \frac{\sum Y_{t,s} \times Ha_{t,s}}{\sum Ha_{t,s}} \tag{1}$$

Where $t$ and $s$ denote the year and political unit, respectively, $Y_{t,s}$ and $Ha_{t,s}$ represent the yield (t/ha) and harvested area (ha). To eliminate non-climatic influences on yield (Anderson et al., 2017a; Iizumi et al., 2014), we calculated expected yields ($Y_{EXt,s}$) by applying a five-year running mean for each state. The percent yield anomaly ($\Delta Y_{t,s}$) relative to expected yield was then computed as follows:

$$\Delta Y_{t,s} = \frac{Y_{t,s} - Y_{EXt,s}}{Y_{EXt,s}} \times 100 \tag{2}$$

Where $t$ and $s$ again represent the year and political unit, respectively. $Y_{t,s}$ represents the yield value (t/ha). Due to the five-year running mean (averaging from $t$-2 to $t$+2), anomalies could not be calculated for 1978, 1979, 2018, and 2019.

To capture the variability of the tropical Indian Ocean SST, we used monthly mean SST grid data at a $1.0° \times 1.0°$ resolution from the Hadley Centre Sea Ice and Sea Surface Temperature dataset (HadISST, Rayner et al., 2003) spanning 1979-2017. The IOB index was defined as the average SST anomaly (SSTA) in the region 20°S-20°N, 40°-110°E, while the Niño 3.4 index was calculated as the SSTA averaged over 5°S-5°N, 170°W-120°W. Before calculating these indices, SST fields were standardized by multiplying each value by the square root of the cosine of latitude.

To assess the impact of meteorological factors on soybean yields in the United States, we selected ten key variables from the Climatic Research Unit (CRU) TS v4.07 dataset and the ERA5 reanalysis (Harris et al., 2020; Hersbach et al., 2020). These include temperature [maximum (Tmx, °C), mean (Tmp, °C), minimum (Tmn, °C), diurnal temperature range (DTR, °C)], precipitation (Precip, mm·d⁻¹), wet day frequency (Wet, days), cloud cover (Cld, %), downward shortwave radiation flux (DSRF, W·m⁻²), root-zone soil moisture (SMroot, m³·m⁻³; Layer 2, 7-28 cm depth), and vapor pressure deficit (VPD, hPa). Eight of these variables were obtained from CRU, which provides monthly mean gridded data at $0.25° \times 0.25°$ resolution, while SMroot was obtained from ERA5 as a proxy for soybean root water uptake. The choice of variables is consistent with previous studies highlighting the role of temperature, precipitation, radiation, soil moisture, and humidity in soybean yields (Gaupp et al., 2020; Gobin and Van de Vyver, 2021; Hamed et al., 2021; Joshi et al., 2021; Leng and Hall, 2019; Ray et al.,

2015; Schauberger et al., 2017), with VPD included as an additional dryness indicator (Ergo et al., 2018). VPD was calculated using the following formulas:

$$e_0 = 6.108\exp\left(\frac{17.27 \times \text{Tmp}}{\text{Tmp}+237.3}\right) \tag{3}$$

$$\text{VPD} = e_0 - e_a \tag{4}$$

where Tmp is the monthly average temperature (°C), and $e_a$ is the average actual vapor pressure (hPa), both from the CRU
dataset. $e_0$ represents the monthly mean saturated vapor pressure (hPa). A summary of all variables and references, including units, is provided in Table S1 in the Supplementary.

To investigate how the IOB mode influences key meteorological factors, we selected monthly values for mean sea level pressure (SLP, Pa), geopotential height at 200 hPa (GPH200, m), and wind components at 925 hPa (meridional, v925 in m·s$^{-1}$, and zonal, u925 in m·s$^{-1}$). All variables were obtained from the ERA5 reanalysis dataset (Hersbach et al., 2020).

**2.2 Statistical analyses**

Before conducting the specific analyses, we employed Gram-Schmidt orthogonalization to remove the linear influence of ENSO (represented by Niño3.4) from the IOB index, other climate indices, meteorological factors, and large-scale circulation fields. This method transforms correlated variables into orthogonal sets by sequentially projecting each target variable onto the space orthogonal to ENSO. The ENSO-independent component of a variable X was calculated as:


$$X_{\perp E} = X - \left(\frac{\langle X, E\rangle}{\langle E, E\rangle}\right) E \tag{5}$$

Where X is the original variable, E the ENSO signal, and $\langle \cdot, \cdot \rangle$ denotes the inner product. Through this procedure, only the variability linearly independent of ENSO is retained, enabling a clearer attribution of Indian Ocean-related effects. We note that while this approach ensures zero-lag statistical independence from ENSO, lead-lag influences cannot be fully eliminated, as ENSO and Indian Ocean warming often co-evolve and interact across seasons. Similar approaches have been applied in
recent climate studies (Hou et al., 2024).

We began by conducting a correlation analysis between multiple climate index time series and U.S. soybean yield data. In this step, we systematically adjusted the time range of each climate index to examine its lagged impact on soybean yield, which allowed us to identify the climate modes with the most significant influence. Next, we applied ridge regression to analyze the relationship between ten meteorological factors and soybean yield, isolating the three factors with the greatest impact on yield
outcomes. For clarity, in this study we define meteorological factors as local surface climate variables that directly affect crop growth (e.g., Tmx, Precip, SMroot, DSRF, and VPD). In contrast, we define atmospheric circulation patterns as large-scale circulation fields that characterize regional and hemispheric variability, including SLP, GPH200, and 925 hPa winds. We then divided the regression analysis into two components: (1) assessing the relationship between climate modes, key meteorological factors, and atmospheric circulation patterns, and (2) examining the link between key meteorological factors and soybean yield.

This framework enabled us to construct an impact chain that connects climate modes, large-scale circulation features, local weather conditions, and ultimately soybean yield outcomes.

## 3 Results

### 3.1 Climate variabilities and soybean yield anomalies

Many studies have focused on the climate variability of the Pacific and Atlantic Oceans, including phenomena such as ENSO
and Tropical Atlantic variability (TAV), which will affect the weather conditions in the United States, resulting in fluctuations in soybean production (Anderson et al., 2019; Qian et al., 2020). However, in addition to these well-studied oceanic drivers, other climatic modes may also exert significant impacts on soybean yields. To explore this further, we conducted a comprehensive analysis of the correlation between soybean yields in key production regions and a broad range of climate indices. These indices include Niño3, Niño4, Niño3.4, Pacific North American Index (PNA), Pacific Decadal Oscillation
(PDO), Southern Oscillation Index (SOI), AMO, Atlantic Meridional Mode (AMM), NAO, Tropical South Atlantic (TSA), Tropical North Atlantic (TNA), Indian Ocean Dipole (IOD), and IOB, among others. See Table S2 in the Supplementary for sources of these indices.

As shown in Fig. 1, during the soybean growth period (MAM-JAS), Niño3.4 and Niño4, which are associated with the ENSO phenomenon, exhibit a statistically significant correlation with U.S. soybean yields at the 95% confidence level based on a
two-tailed t-test. This finding aligns with previous studies, which have well-documented the influence of ENSO on soybean production (Anderson et al., 2017b; Hamed et al., 2023).

However, our analysis extends beyond the growing season to consider climate variability during the winter months prior to soybean planting. We find that the IOB mode, which captures large-scale warming or cooling in the Indian Ocean, plays a substantial role in determining soybean yields before sowing. The year-to-year anomalies in soybean yield exhibit the strongest
Pearson correlation (-0.41) with the IOB index during ND(-1)J (November and December of the year preceding harvest and January of the harvest year), which was identified as the optimal 3-month window following an exhaustive correlation screening across all possible periods; this relationship is statistically significant at the 99% confidence level (two-tailed t-test; Fig. 1). In a simple linear regression framework, this corresponds to a coefficient of determination of $R^2 = 0.1681 (\approx 16.8\%)$, indicating that approximately 16.8% of the interannual variability in soybean yield is explained by the IOB mode during this
pre-growing season window.

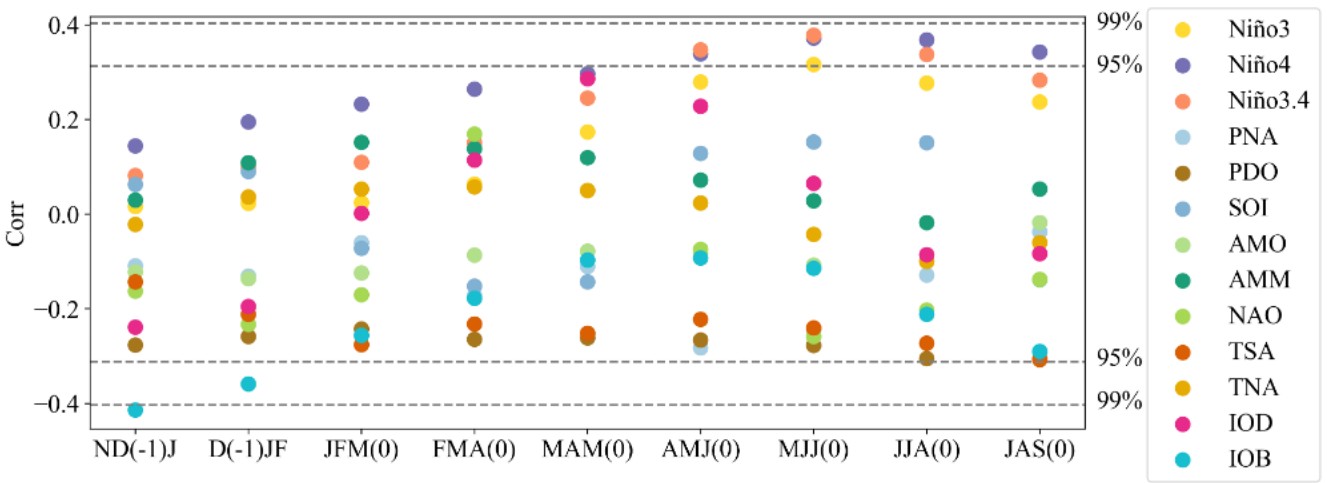

**Figure 1: Correlation between the United States soybean yield anomalies and climate indices over 3-month periods from 1980 to 2017. The horizontal axis shows each 3-month window, and the vertical axis shows Pearson correlation coefficients. Colors represent different climate indices as listed in the legend. Dashed lines mark the vertical axis at the 95% and 99% significance levels based on the t-test.**

Figure 2(a) illustrates the spatial distribution of soybean production in the United States. It is evident that soybean production is concentrated in the central United States, with Illinois, Iowa, Minnesota, Indiana, and Nebraska having the highest yields, accounting for more than 50% of the total soybean production in the country.

To explore how soybean yields respond to variations in the IOB mode across different U.S. regions, we conducted a regression analysis. The findings reveal that the IOB mode during ND(-1)J exerts a significant negative influence on soybean yields, with an inverse relationship between the IOB index and yields observed in most states (Fig. 2(b)). This negative correlation was especially pronounced in several key agricultural states. Notably, South Dakota, Kansas, Missouri, Illinois, Michigan, Indiana, Ohio, Kentucky, Tennessee, and New York displayed significant sensitivities to fluctuations in the IOB mode. The relationship between the IOB index and soybean yields in these regions passed the 90% significance threshold, indicating that the impact of the IOB mode on soybean production in these states is both statistically robust and consistent.

To quantify the overall impact of IOB fluctuations on U.S. soybean production, we calculated a weighted sum of the regression coefficients, with the weights based on the average harvested area and expected yields across individual regions. This approach provides a more comprehensive assessment of the IOB mode's effect on national production. The findings indicate that an increase in the IOB index is associated with a decrease in soybean production in nearly all regions of the United States (Fig. 2(c)). Specifically, a one standard deviation increase in the IOB index during ND(-1)J corresponds to a yield reduction of approximately 4.7 million tonnes in regions experiencing significant declines, while regions with notable yield increases were virtually nonexistent. Among these regions, Illinois saw the largest decline, with an estimated loss of around 800,000 tonnes. In aggregate, an increase of one standard deviation in the IOB index during ND(-1)J corresponds to a reduction of approximately 4.0% in total soybean production across the United States.

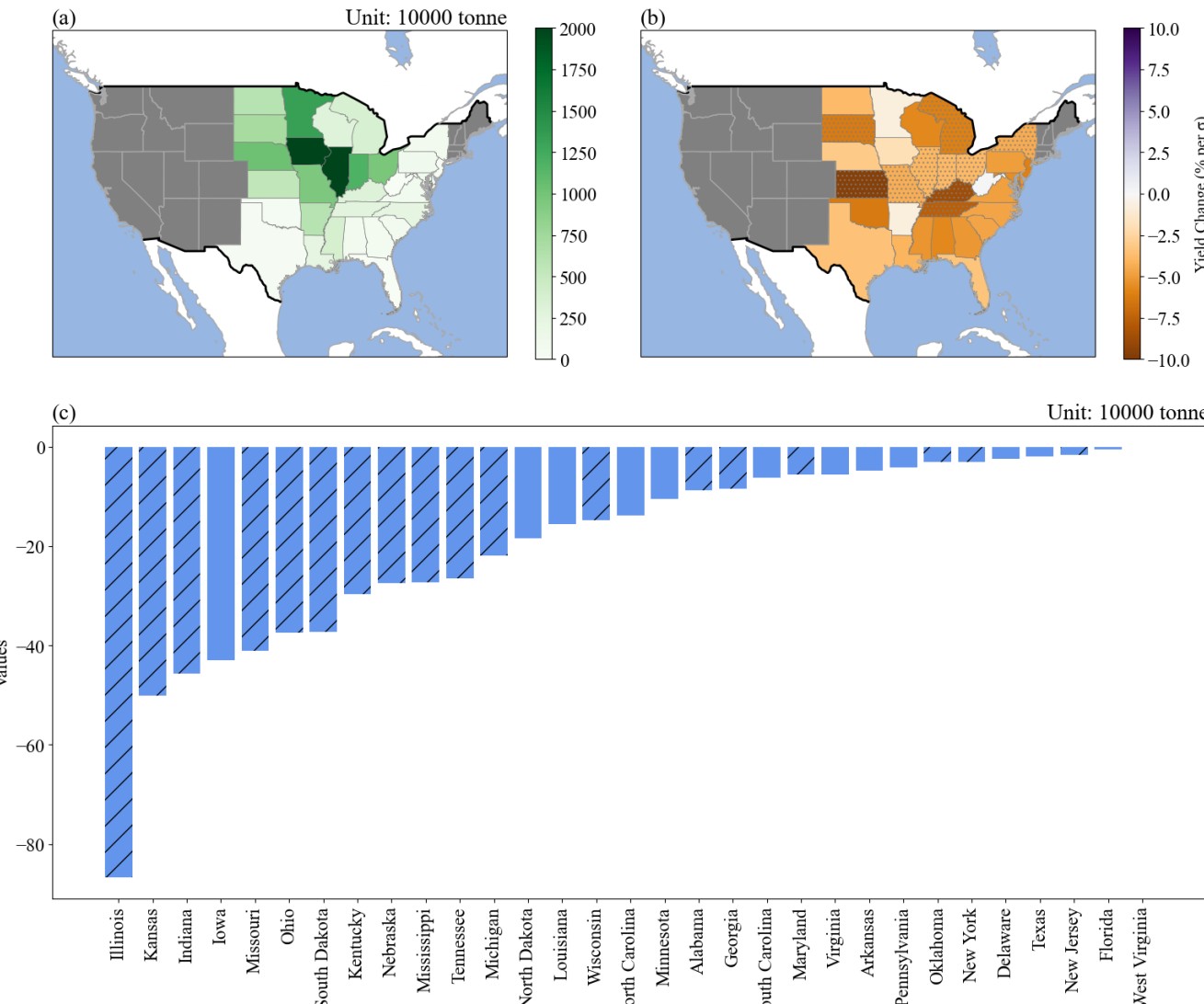

Figure 2: Spatial distribution of (a) average soybean production, (b) sensitivity of soybean yield anomalies (%), and (c) sensitivity of total soybean production (10,000 tonnes) to the IOB index during ND(-1)J from 1980 to 2017. Panel (a) shows the multi-year average production in each state. Panels (b) and (c) show the change in the percent yield anomaly and total production, respectively, per one standard deviation (1σ) increase in the standardized IOB index. The IOB index was standardized before analysis. Dots in (b) and striped bars in (c) indicate states where the correlation with the IOB index is statistically significant at the 90% confidence level based on the t-test.

## 3.2 Impacts of weather conditions on soybean yields

To identify the key factors influencing soybean yields in the United States, we considered ten climate indicators (Tmp, Tmx, Tmn, DTR, DSRF, Precip, VPD, Cld, wet, and SMroot). Given the interdependence among these predictors, ridge regression was employed to stabilize coefficient estimates by penalizing multicollinearity. The regression results indicate that diurnal temperature range (DTR) during the reproductive stage (JAS) exerts the largest influence on yield anomalies, followed by root-zone soil moisture (SMroot) and maximum temperature (Tmx) (Table S3 in the Supplementary).

To more precisely assess the influence of these meteorological factors on soybean yields in the United States, we conducted a regression analysis examining their relationships with soybean yield across various states. Figure 3(a) demonstrates that an increase in the DTR is consistently associated with a decrease in soybean yields across nearly all regions of the United States. Statistically significant reductions in yields are observed in approximately 58% of the regions (P < 0.1), indicating that large temperature swings between day and night negatively affect soybean productivity. Figure 3(b) indicates a positive correlation between SMroot and soybean yields in most areas, particularly in the Midwest and Southeast, where significant positive correlations (P < 0.1) are evident. We also find that soybean yields are significantly impacted by DTR in areas similar to those affected by SMroot, suggesting that the effects of DTR are particularly pronounced in regions with limited water supply. Figure 3(c) illustrates that higher maximum temperatures (Tmx) are linked to decreased soybean yields in the southern and central United States, with significant negative correlations observed in these areas (P < 0.1). Notably, the spatial patterns of DTR and Tmx closely resemble each other, both showing an inverse relationship with SMroot.

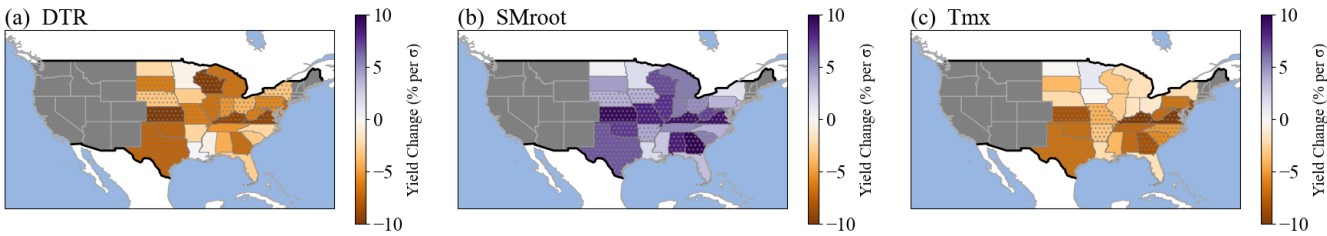

**Figure 3: Sensitivity of soybean yield anomalies to (a) diurnal temperature range (DTR, °C), (b) root zone soil moisture (SMroot, $m^3 \cdot m^{-3}$), and (c) maximum temperature (Tmx, °C) during the growing season. Meteorological variables in this figure were standardized by removing the mean and dividing by the standard deviation before analysis. Values indicate the change in the percent yield anomaly (%) per one standard deviation (1σ) increase in each standardized variable. Dots denote correlations significant at the 90% confidence level (t-test).**

## 3.3 Soil Moisture Memory Effect

During ND(-1)J, an increase in the IOB index results in a widespread reduction in SMroot across the United States. Significant soil moisture anomalies (P < 0.1) are particularly evident in North Dakota, South Dakota, Minnesota, Illinois, Georgia, South Carolina, North Carolina, and parts of Nebraska, Kansas, Iowa, Missouri, and Indiana (Fig. 4(e)). This decrease in SMroot

appears to be associated with anomalies in Precip and VPD, as regions experiencing reduced SMroot generally align with areas of increased VPD and decreased Precip.

As illustrated in Fig. 4(a), the warming signals from the Indian Ocean can affect the United States across the Pacific. The warming of the Indian Ocean enhances convective activities and upward motion, leading to the development of negative outgoing longwave radiation (OLR) anomalies. These anomalies generate Rossby waves that propagate from the tropical

Indian Ocean to the United States via East Asia and the North Pacific (Hu et al., 2023; Ratnam et al., 2012). The response pattern of 200-hPa geopotential heights consists of a low-high system over the United States (Fig. 4(a)). This leads to southeasterly wind anomalies, which bring warmer and wetter equatorial air to the United States, causing widespread temperature increases in the United States and more precipitation in the southeastern United States (Figs. 4(b) and 4(c)). However, the southeasterly winds that bring warm and moist air shift to southwesterly winds in the northern United States

(Fig. 4(a)), and the northeasterly wind anomaly transfers drier air from the polar regions to the Northern U.S. (Wang et al., 2013). Concurrently, the north-central United States becomes a water vapor divergence area (Fig. S3 in the Supplementary), reducing Precip and contributing to decreased SMroot in that area (Figs. 4(b) and 4(e)). In the southeastern United States, although Precip increases, the concurrent rise in Tmp intensifies VPD, contributing to the observed decrease in SMroot (Figs. 4(c)-(e)). Elevated Tmp could amplify VPD by accelerating soil and plant moisture loss, as shown in previous studies (Yin et

al., 2014; Zhao et al., 2023).

For the root zone, soil moisture memory ranges from several days to up to a year and shows a long memory for dry soil moisture regimes (Stacke and Hagemann, 2016). In this study, we found that soil moisture anomalies during ND(-1)J are significantly positively correlated with soil moisture anomalies in JAS (Supplementary Fig. S3). Specifically, the strongest positive correlations are observed over the U.S. Midwest and Northern Plains, which represent the core soybean production

regions, indicating that early-season soil moisture conditions in these areas persist into the summer. This "memory effect" suggests that reductions in ND(-1)J soil moisture can influence root-zone moisture during the reproductive stage, thereby affecting soybean yields and highlighting the role of early-season hydrological conditions in modulating summer drought risk.

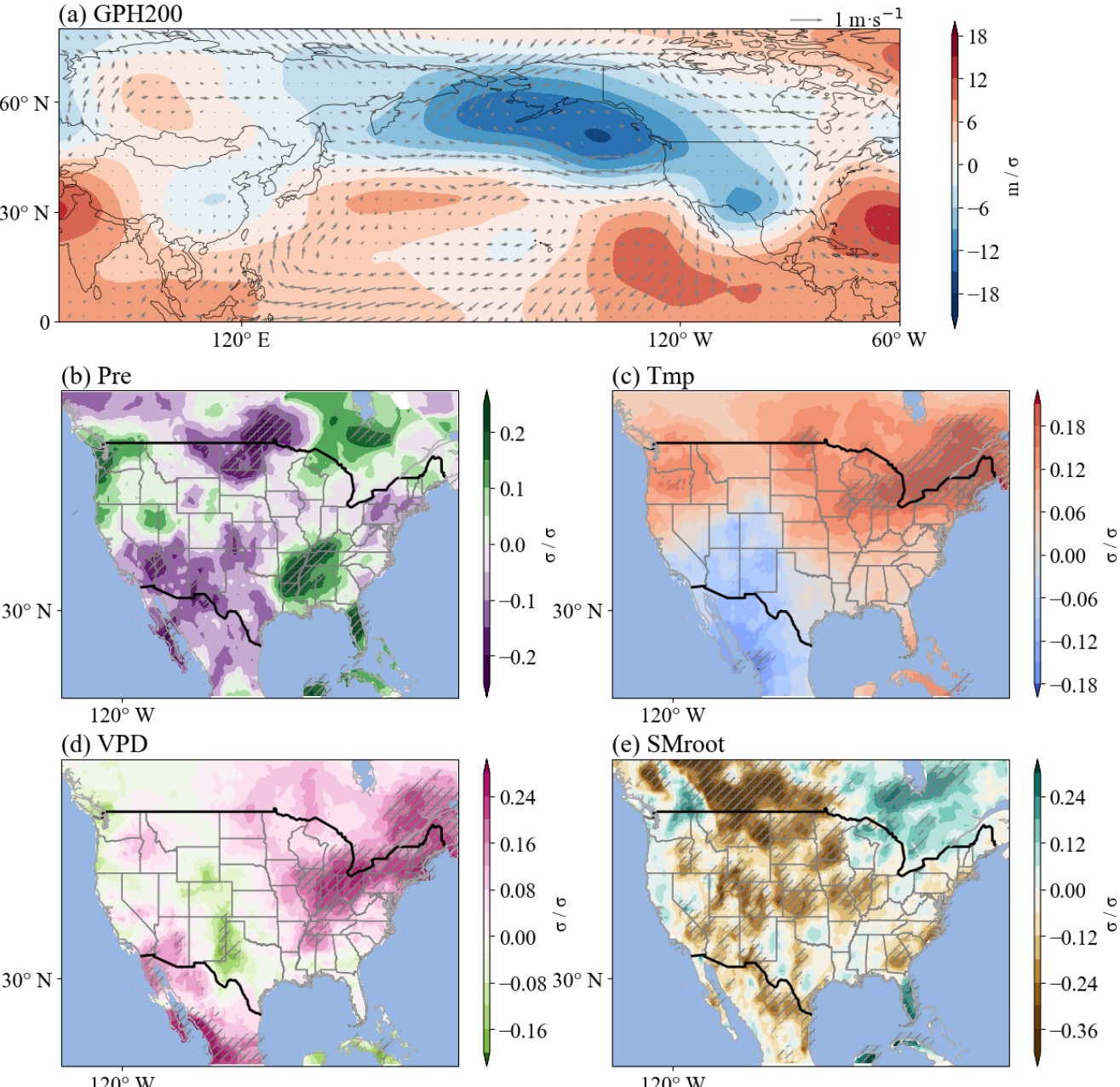

Figure 4: Responses of (a) 200 hPa geopotential height (GPH200, m), (b) precipitation (Precip, mm·d⁻¹), (c) mean temperature (Tmp, °C), (d) vapor pressure deficit (VPD, hPa), and (e) root zone soil moisture (SMroot, m³·m⁻³) to the IOB index during ND(-1)J. Vectors in (a) show wind responses at 925 hPa (m·s⁻¹) to a 1σ increase in the standardized IOB index. All variables except GPH200 and winds were standardized before analysis. Color bar values in (a) represent changes in GPH200 (m) per 1σ increase in the standardized IOB index, while color bar values in (b-e) represent standardized changes (in σ units) in meteorological variables per 1σ increase in the standardized IOB index. Shaded areas with diagonal hatching indicate regions where the response is statistically significant at the 90% confidence level (t-test).

### 3.4 Sea Surface Temperature Feedback Effect

The warming of the Indian Ocean during the ND(-1)J could indirectly influence JAS meteorological conditions in the United States by affecting Pacific SSTs. It is associated with a Matsuno-Gill-type response, which excites atmospheric Kelvin waves propagating eastward (Gill, 1980; Matsuno, 1966). This results in easterly wind anomalies over the western-central equatorial Pacific. Strengthened easterly winds lead to a shallower thermocline, which brings colder waters closer to the surface, contributing to the cooling of SSTs in the eastern Pacific. This cooling effect can persist throughout the summer months, as illustrated in Fig. 5(a), and is a key component of the development of La Niña events (Cai et al., 2019; Wang, 2019; Wu and Kirtman, 2004; Xie et al., 2016). During La Niña events, there are significant atmospheric circulation anomalies that further impact the United States. For example, Fig. 5(b) shows that during La Niña years, 200 hPa geopotential height anomalies are predominantly positive in the mid-latitudes, while negative anomalies occur in the tropics and certain high-latitude regions (Luo and Lau, 2020). These geopotential height anomalies reflect the establishment of an anomalous high-pressure system over much of the United States.

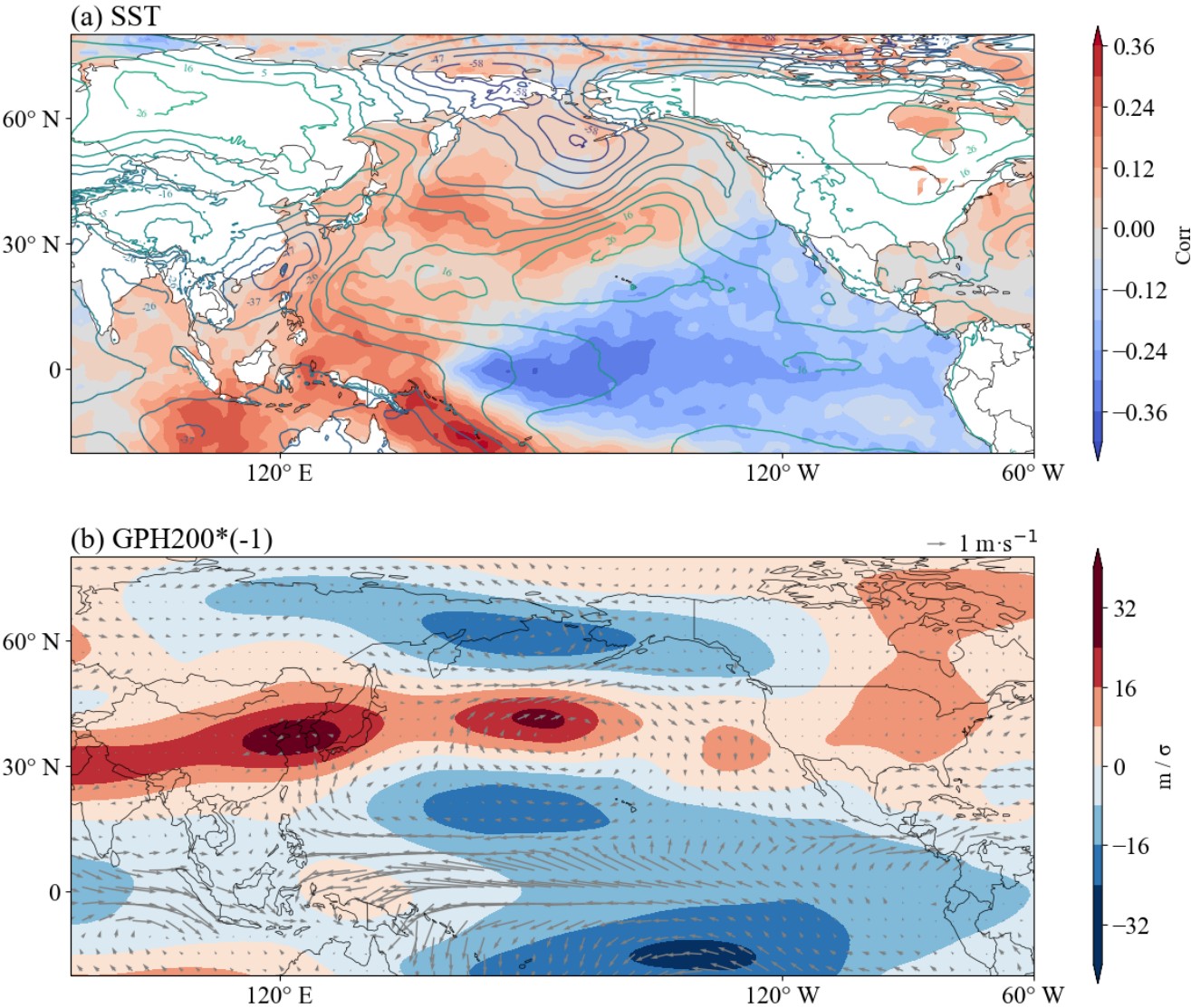

**Figure 5: (a) Correlation between sea surface temperature (SST, °C) and the IOB index, and (b) responses of 200 hPa geopotential height anomalies (GPH200, m) to the Niño3.4 index during JAS. Solid contours in (a) represent sea level pressure (SLP) anomalies (units: Pa). Vectors in (a) show wind responses at 925 hPa (m·s⁻¹) to a 1σ decrease in the standardized IOB index. GPH200, winds, and SLP are shown in original units; the IOB and Niño3.4 indices were standardized before analysis. Color bar values in (a)**
260 **represent correlation coefficients; color bar values in (b) represent the change in GPH200 (m) per 1σ decrease in the standardized Niño3.4 index.**

Such conditions lead to changes in meteorological patterns across the United States, including warmer temperature anomalies and decreased precipitation (Jia et al., 2016; Jong et al., 2020, 2021; Mo et al., 2009; Wang et al., 2007). Consequently,
substantial reductions in Precip and increases in Tmx are shown in Figs. 6(a) and 6(b). The elevated Tmx exacerbates evapotranspiration, driving higher moisture loss from both the soil and the plants. This warming effect intensifies the drought-

like conditions across several regions of the United States. Additionally, a noticeable rise in the VPD (Fig. 6(c)) further compounds the stress on crops by increasing the demand for moisture in the atmosphere. This heightened atmospheric moisture deficit, combined with reduced precipitation, directly leads to declining SMroot levels (Fig. 6(d)). These patterns indicate a

clear decline in soil water availability, heightening the risk of drought conditions across affected regions. Moreover, Fig. 6(f) reveals a significant increase in the DTR across soybean-growing regions. This increase is likely associated with a reduction in Cld (Fig. 6(e)), which allows for more extreme temperature variations between day and night. The pattern observed in this study is similar to those examined by Doan et al. (2022).

These results indicate that the variability of the IOB can affect the meteorological elements in the United States by influencing

the development of ENSO. The hot and dry conditions resulting from these meteorological changes are particularly detrimental to soybean yields, as highlighted by Anderson et al. (2017a)and Hamed et al. (2023). The combined impacts of increased Tmx and DTR, decreased Precip, elevated VPD, and reduced SMroot create an environment highly unfavorable for soybean production. This highlights the critical role of these climate drivers in influencing agricultural outcomes under positive IOB conditions (Fig. S5 in the Supplementary).

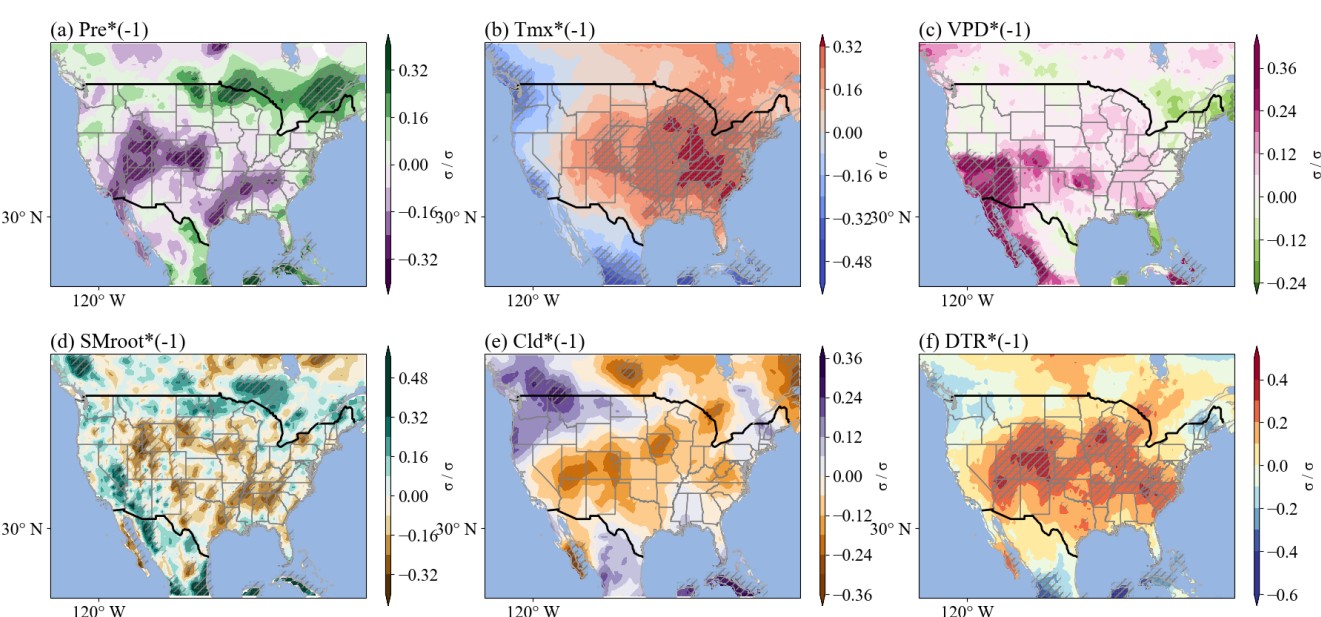

**Figure 6:Responses of (a) precipitation (Precip, mm·d⁻¹), (b) maximum temperature (Tmx, °C), (c) vapor pressure deficit (VPD, hPa), (d) root zone soil moisture (SMroot, m³·m⁻³), (e) cloud cover (Cld, %), and (f) diurnal temperature range (DTR, °C) to the Niño3.4 index during JAS. All variables were standardized before analysis. Values indicate the standardized change (in σ units) in**
**meteorological variables per 1σ decrease in the standardized Niño3.4 index. Shaded areas with diagonal hatching indicate regions where the response is statistically significant at the 90% confidence level (t-test).**

## 4 Summary and discussion

This study investigates the significant influence of Indian Ocean basin-wide SST anomalies on U.S. soybean yields. We show that the IOB mode during ND(-1)J explains approximately 16% of the interannual variability in soybean yields. The warming of the Indian Ocean influences U.S. climate through two pathways (Fig. 7). First, it excites Rossby wave trains that alter atmospheric circulation, leading to reductions in U.S. soil moisture as early as ND(-1)J. Second, it is associated with a Matsuno-Gill-type response, resulting in lower sea surface temperatures in the eastern equatorial Pacific. This response persists into boreal summer, intensifying hot-dry anomalies across most of the United States during JAS. Together, these mechanisms create unfavorable hydroclimatic conditions during the soybean reproductive stage, thereby amplifying risks of widespread yield loss.

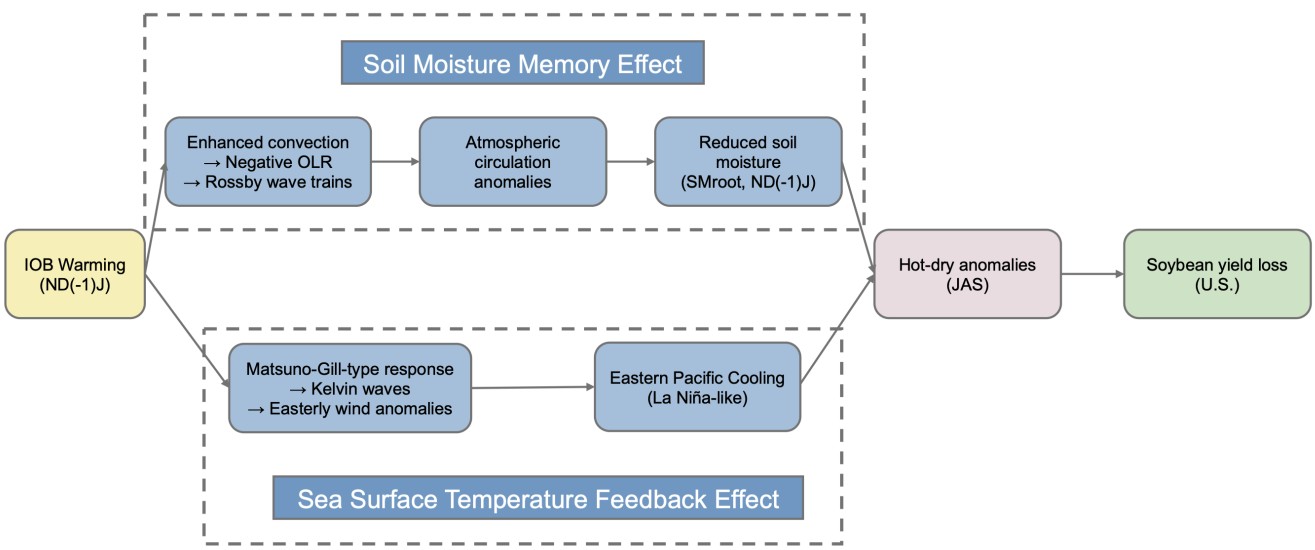

**Figure 7: Schematic illustration of the mechanisms linking IOB warming during ND(-1)J to U.S. soybean yield loss.**

Previous studies have predicted crop yield based on early signals of ENSO and explained the lagged impact of that on crop yield through soil moisture memory and sea surface temperature memory (Anderson et al., 2017b; Cao et al., 2023; Hamed et al., 2023; Von Bloh et al., 2023). While ENSO is a major driver of global climate, its interactions with other climate systems (Cai et al., 2019; Fan and Meng, 2023; Strong et al., 2020; Zhang et al., 2022), like the Indian Ocean, have been overlooked. This study is the first to highlight the significant impact of the IOB index on the United States soybean yields, identifying the IOB as a key predictive factor with a nine-month lead time. Our findings offer early warnings of potential yield reductions, enabling farmers to adapt management practices and guiding governments in adjusting trade policies to mitigate climate-related risks.

Ridge regression identifies Tmax and SMroot as the most influential meteorological variables for soybean yield anomalies. Tmax primarily reflects atmospheric heat stress, while SMroot measures subsurface water availability and drought persistence.

These variables are physically interdependent: dry soils reduce evaporative cooling, increasing Tmax, while higher Tmax enhances evapotranspiration, further reducing SMroot and creating a reinforcing feedback (Miralles et al., 2014). Despite this interdependence, ridge regression mitigates multicollinearity through coefficient shrinkage, allowing both variables to retain meaningful contributions and jointly characterize compound climate risks. Many other climate variables affecting crop growth also interact, such as soil moisture, precipitation, and temperature, which together determine the water and heat conditions

experienced by crops. These complex interactions make yield variability difficult to attribute to any single factor and highlight the importance of modeling approaches that explicitly capture interacting and reinforcing climatic influences on crop production.

DTR also emerged as a critical predictor in our analysis, with higher DTR associated with reduced soybean yields. Larger DTR typically reflects higher daytime temperatures combined with cooler nights, which can increase evapotranspiration and

320 atmospheric moisture demand or reduce photosynthetic rates, thereby exacerbating water stress and inhibiting growth (Hatfield et al., 2011; Lobell, 2007). In some regions, increases in DTR can also intensify drought and worsen crop water deficits(Feng et al., 2025). Several studies have shown that elevated DTR during summer is associated with reduced soybean yields(Goulart et al., 2021; Verón et al., 2015). Our analysis further highlights the significant role of DTR in contributing to soybean yield variability, suggesting it should be incorporated into climate-crop impact assessments and seasonal prediction frameworks.

This study employed a lead-lag correlation and regression framework to quantify the influence of preseason climate variability on U.S. soybean yields. This approach captures both concurrent and delayed responses of crop yields to large-scale ocean-atmosphere anomalies, allowing for the identification of early predictors of agricultural risk. Although lead-lag techniques are well established in climate research (Hou et al., 2024; Yang and Xing, 2022; Yang et al., 2025), they have been less frequently applied to quantify climate-agriculture linkages, offering new perspectives on the temporal pathways through which climate

variability affects crop production. Furthermore, this framework may be useful for investigating compound and multi-risk events, where multiple climatic drivers interact or occur sequentially. By tracing how early-season anomalies evolve and combine to influence subsequent impacts, it provides a systematic means to explore interconnected climate processes relevant to agricultural and environmental risk assessments.

There are several limitations to this study. Our analysis is based on 38 years of agricultural and meteorological records, which

may constrain robustness and introduce uncertainty, particularly in regions with sparse observations. Varying spatial scales and mismatches between crop statistics and climate datasets can further affect accuracy. In addition to commonly used climate indicators such as precipitation and temperature, which have been widely applied in previous studies (Lobell, 2007; Mourtzinis et al., 2015; Ray et al., 2015), more specific agroclimatic indices,   for example, heat-stress days, evapotranspiration deficits, and phenology-specific metrics, could improve yield prediction skill and attribution. Crop yield variability is driven by multiple

interacting climate modes, so combined analyses that account for these interactions are needed. Finally, non-climatic factors, including management practices and genetic improvements, were not explicitly represented here and should be integrated into future modeling efforts to reduce uncertainty and enhance the applicability of yield predictions.

Although our study focuses on the interannual variability of the IOB index, its broader implications under climate change warrant attention. Observations over recent decades show a persistent warming of the tropical Indian Ocean, and climate model projections indicate further, spatially heterogeneous warming in the future, with stronger signals in certain regions such as the western equatorial basin and Arabian Sea (Cai et al., 2019; Gopika et al., 2025; Rao et al., 2012; Sharma et al., 2023). Such background warming may alter both the amplitude and frequency of IOB variability, potentially intensifying its teleconnections with the Pacific and North America. At the same time, U.S. soybean yields are projected to decline substantially under climate change. For example, Hultgren et al. (2025) suggest large reductions under high-emission scenarios, while Schlenker and Roberts (2009) estimate a 30-40% decline even under the lowest-emission scenario by the end of the century. Moreover, climate change is expected to exacerbate temperature extremes and drought severity, leading to greater variability in soybean yields (Proctor et al., 2025). These parallel trends raise the possibility of compounding interactions between enhanced IOB variability and an increasingly drought-prone U.S. climate (Schlenker and Roberts, 2009). Future research should therefore employ coupled climate-crop modeling to assess how long-term ocean warming interacts with interannual IOB signals, providing insights into adaptation strategies such as breeding stress-tolerant cultivars, optimizing planting schedules, and improving water management. Integrating IOB monitoring into agricultural early-warning systems would further enhance resilience to future climate risks.

In summary, this study demonstrates that Indian Ocean SST variability is a powerful but underutilized source of predictive information for U.S. soybean yield risk assessment. By linking preseason IOB anomalies to summer weather extremes and yield losses, this work deepens understanding of the ocean-atmosphere-agriculture nexus, offering opportunities to enhance seasonal forecasting, strengthen supply chain planning, and inform international trade strategies in an era of increasing climate uncertainty.

**Data availability**

The U.S. State-level soybean harvest data were obtained from the USDA's NASS Quick Stats database (https://quickstats.nass.usda.gov/). The crop planting and harvest dates were obtained from the Crop Calendar Dataset (Sacks et al., 2010). Monthly mean SST grid data from the Hadley Centre Sea Ice and Sea Surface Temperature dataset (Rayner et al., 2003). Monthly gridded maximum temperature, mean temperature, minimum temperature, diurnal temperature range, mean precipitation, surface downward short-wave radiation flux, wet day frequency, vapor pressure, and cloud cover were obtained from the Climatic Research Unit (CRU) v4.07 dataset (Harris et al., 2020). The monthly root zone soil moisture, geopotential height at 200 hPa, wind components at 925 hPa, and mean sea level pressure is publicly available from the European Centre for Medium-Range Weather Forecasts dataset (ERA5, Hersbach et al., 2020).

The code used for data integration and analysis in this study is publicly available in the GitHub repository "The-Preseason-Warming-of-the-Indian-Ocean-Resulting-in-Soybean-Failure-in-U.S." at https://github.com/MenghanLi-Maian/The-Preseason-Warming-of-the-Indian-Ocean-Resulting-in-Soybean-Failure-in-U.S./releases/tag/v1.0.0.

## Author contribution

Menghan Li was involved in data curation, formal analysis, investigation, methodology, visualization, writing-original draft, and writing-review and editing. Yi Zhou was involved in writing-review and editing. Yurong Hou was involved in Validation. Xichen Li was involved in conceptualization, supervision, and writing-review and editing.

## Competing interests

The authors declare no conflict of interest relevant to this study.

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
