# Peer review of "The Preseason Warming of the Indian Ocean Resulting in Soybean Failure in U.S."

_EGUsphere, 2025_

## Author Comment (AC2)

**Response to reviewer 2**

Thank you for carefully reviewing our manuscript. We address each point below, quoting the comments and providing our responses. All changes to the manuscript are indicated with section/line references.

Methods:

1. **Reviewer Comment:** More details are needed on the Gram-Schmidt orthogonalisation method, it is too vague for now. Among questions and points that I would like to see explained: Can you explain what it does and how you applied it? Can you expand the motivation for this (or what would happen without this step)? Could you be missing signal or information by doing that (special attention to ENSO here)? And if this is a common approach / which other studies have done this before?

**Response:** We have substantially expanded the explanation of Gram–Schmidt orthogonalization in the Statistical analyses (Section 2.2). Specifically, Gram–Schmidt orthogonalization is a standard technique in linear algebra that provides a straightforward framework for transforming a set of potentially correlated variables into an orthogonal (uncorrelated) set by sequentially projecting each variable onto the orthogonal space of the previously processed ones (Giraud et al., 2005). This approach was applied to remove the linear ENSO (Niño 3.4) signal from our climate indices before further analyses. It ensures that subsequent correlation and regression analyses isolate the Indian Ocean effects independently of ENSO.

We acknowledge that Gram-Schmidt orthogonalization has potential limitations, such as its dependence on the ordering of variables and sensitivity to numerical instability in the presence of multicollinearity. However, since we only orthogonalized ENSO, the ordering issue is minimized. Nevertheless, we note that the method guarantees independence only at zero lag, so lead-lag interactions between ENSO and Indian Ocean warming may not be fully removed. To establish precedent, we also refer to recent studies that adopted this approach to control for ENSO influences in climate analyses (Hou et al., 2024).

**Manuscript changes:** In Section 2.2 (Lines 97–99), we revised the text as follows:

B Before conducting the specific analyses, we employed Gram–Schmidt orthogonalization to remove the linear influence of ENSO (represented by Niño3.4) from the IOB index, other climate indices, meteorological factors, and large-scale circulation fields. This method transforms correlated variables into orthogonal sets by sequentially projecting each target variable onto the space orthogonal to ENSO. The ENSO-independent component of a variable $X$ was calculated as:

$$X_{\perp E} = X - \left( \frac{\langle X, E \rangle}{\langle E, E \rangle} \right) E \tag{5}$$

Where $X$ is the original variable, $E$ the ENSO signal, and $\langle \cdot, \cdot \rangle$ denotes the inner product. Through this procedure, only the variability linearly independent of ENSO is retained, enabling a clearer attribution of Indian Ocean–related effects. We note that while this approach ensures zero-lag statistical independence from ENSO, lead–lag influences cannot be fully eliminated, as ENSO and Indian Ocean warming often co-evolve and interact across seasons. Similar approaches have been applied in recent climate studies (Hou et al., 2024).

2. **Reviewer Comment:** Can you explain and justify the initial choice of meteorological variables? Is this based on previous studies, do similar studies select the same variables? It reads a bit unclear and arbitrary right now.

**Response:** The meteorological variables were selected based on previous agronomic and climate-yield studies that consistently highlight temperature (including its diurnal range, DTR), precipitation, radiation, soil moisture, and humidity as the dominant drivers of soybean yield variability (Gaupp et al., 2020; Hamed et al., 2021; Joshi et al., 2021; Otkin et al., 2016; Ray et al., 2015; Schauberger et al., 2017). In addition, vapor pressure deficit (VPD) has been widely used as a proxy for atmospheric dryness and crop stress (Ergo et al., 2018). To provide transparency, we now include a summary of all variables, their definitions, sources, and supporting references in the Supplementary Material (Table S1).

We also note that, in addition to the widely recognized variables, we included cloud cover (Cld) as an exploratory factor. This variable is less commonly studied in soybean yield analyses, but we considered it relevant due to its potential to affect surface energy balance and crop growth. This rationale is now clarified in the revised manuscript and Supplementary Table S1.

**Manuscript changes:** In Section 2.1 (Lines 81–92), we rewrote the paragraph as:

To assess the impact of meteorological factors on soybean yields in the United States, we selected ten key variables from the Climatic Research Unit (CRU) TS v4.07 dataset and the ERA5 reanalysis (Harris et al., 2020; Hersbach et al., 2020). These include temperature [maximum (Tmx, °C), mean (Tmp, °C), minimum (Tmn, °C), diurnal temperature range (DTR, °C)], precipitation (Pre, mm·d⁻¹), wet day frequency (Wet, days), cloud cover (Cld, %), downward shortwave radiation flux (DSRF, W·m⁻²), root-zone soil moisture (SMroot, m³·m⁻³; Layer 2, 7–28 cm depth), and vapor pressure deficit (VPD, hPa). Eight of these variables were obtained from CRU, which provides monthly mean gridded data at 0.25° × 0.25° resolution, while SMroot was obtained from ERA5 as a proxy for soybean root water uptake. The choice of variables is consistent with previous studies highlighting the role of temperature, precipitation, radiation, soil moisture, and humidity in soybean yields (Gaupp et al., 2020; Gobin and Van de Vyver, 2021; Hamed et al., 2021; Joshi et al., 2021; Leng and Hall, 2019; Ray et al., 2015; Schauberger et al., 2017), with VPD included as an additional dryness indicator (Ergo et al., 2018). VPD was calculated using the following formulas:

$$e_0 = 6.108\exp\left(\frac{17.27 \times \text{Tmp}}{\text{Tmp}+237.3}\right) \qquad (3)$$

$$\text{VPD} = e_0 - e_a \qquad (4)$$

Where Tmp is the monthly average temperature (°C), and $e_a$ is the average actual vapor pressure (hPa), both from the CRU dataset. $e_0$ represents the monthly mean saturated vapor pressure (hPa). A summary of all variables and references, including units, is provided in Table S1 in the Supplementary.

**3. Reviewer Comment:** It is not clear in the text to me how root zone soil moisture is obtained or calculated. You refer to the ERA5 dataset, but as far as I am aware, this variable is not available on the ERA5 repository.

**Response:** In the original manuscript, we referred to "root-zone soil moisture" (SMroot) but did not provide sufficient detail. SMroot was directly obtained from the ERA5 reanalysis dataset as the volumetric soil water content ($m^3 \cdot m^{-3}$). ERA5 provides soil moisture for four layers (0–7 cm, 7–28 cm, 28–100 cm, and 100–289 cm). For this study, we used Layer 2 (7–28 cm depth), which corresponds to the major root water uptake zone for soybean crops. Previous agronomic studies indicate that soybean roots extract most water from the top 30 cm of soil, especially during the reproductive phase, making Layer 2 a reasonable proxy for root-zone soil moisture(Fan et al., 2016; Zhang et al., 2024).

**Manuscript changes:** In Section 2.1 (Lines 85–87), we rewrote the soil moisture description as: "In addition, root zone soil moisture (SMroot, $m^3 \cdot m^{-3}$) was obtained from the ERA5 reanalysis dataset(Hersbach et al., 2020), using Layer 2 (7–28 cm depth) as a proxy for soybean root water uptake."

**4. Reviewer Comment:** When comparing IOB with meteorological variables, you extract SLP from CRU but geopotential height at 200 hPa, and wind components at 925 hPa from ERA5. ERA5 also has SLP, so is there a reason for this? I would argue that having all variables from the same source would guarantee consistency. If you decide to keep SLP from CRU, it should be shown how similar it behaves between the two sources.

**Response:** We thank the reviewer for carefully checking this point. We would like to clarify that in our analysis, SLP was in fact obtained from ERA5, not from CRU. The reference to CRU in the Methods was a writing error. In the revised manuscript, we have corrected this and now state explicitly that all circulation variables (SLP, GPH200, and wind components) were consistently obtained from ERA5 (Section 2.1). We apologize for the oversight and thank the reviewer for helping us improve the clarity of the manuscript.

**Manuscript changes:** We corrected the description of the data source in Section 2.1 (line 95). The sentence has been revised to: "All variables were obtained from the ERA5 reanalysis dataset (Hersbach et al., 2020)."

5. **Reviewer Comment:** The last paragraph of the section 2.2 is confusing. On line 104, number (1), you distinguish between meteorological factors and atmospheric circulation patterns? What exactly do you refer to when you mention atmospheric circulation patterns, this has not been introduced before. Would this be the SLP, GPH200 and the wind components? If so, SLP is not an atmospheric circulation variable, and needs to be corrected. If not, then it would need to be better explained or rewritten to improve clarity.

**Response:** We agree that our terminology was not sufficiently clear in the original manuscript. In the revised manuscript, we have clarified this terminology. Specifically, we now explicitly define:

Meteorological factors as local surface climate variables that directly affect crop growth (e.g., temperature, precipitation, soil moisture, radiation, and humidity).

Atmospheric circulation patterns as large-scale circulation fields that characterize regional and hemispheric circulation variability (e.g., sea-level pressure, geopotential height, and winds).

Although we acknowledge that sea-level pressure (SLP) is sometimes grouped as a surface variable, in this study we treat SLP as part of the large-scale circulation fields because it reflects broad-scale pressure systems and circulation anomalies. This distinction is now clearly stated in the Methods section to avoid confusion.

**Manuscript changes:** In Section 2.2, we added a sentence: "For clarity, in this study, we define meteorological factors as local surface climate variables that directly affect crop growth (e.g., Tmx, Pre, SMroot, DSRF, and VPD). In contrast, we define atmospheric circulation patterns as large-scale circulation fields that characterize regional and hemispheric variability, including SLP, GPH200, and 925 hPa winds."

Results & Discussion:

1. **Reviewer Comment:** The results section combines both actual results and contextualisation aspects that should go into the discussion. And as a consequence, the discussion section is rather small and underdeveloped, looking more like a conclusion than a discussion. Based on that, I would suggest to have the discussion considerably expanded, with the main findings properly contextualised there. For example, the authors find DTR to be important for soybean yield using the ridge regression, which is a statistical approach. I'd like to see potential physical explanations for that (after all, DTR is the difference between two other variables, which could mean many things). Also, have other

studies found similar or diverging relations between DTR and soybean yields in the area of study? These aspects should be properly discussed (you could move some of the small contextualisation points from the results to the discussion and expand them there into a coherent text).

**Response:** We have extensively revised the Discussion. We expanded it to include:

(1) A detailed interpretation of the meteorological predictors Tmax and SMroot, explaining their complementary roles in capturing atmospheric heat stress, soil water availability, and nighttime temperature effects.

(2) A discussion of the importance of DTR for soybean yield assessments.

These additions strengthen the physical and agronomic interpretation of our results.

2. **Reviewer Comment:** I also missed the theoretical implications of the findings: what does it mean to have IOB index influencing soybean variability (beyond the practical point of using it to monitor it in advance)? For instance, can it have any interactions with other major climate phenomena, such as climate change? While this is not the focus of the paper, it could still be briefly discussed. Ex: *What are the future projections for the IOB index? What are the future projections for soybean production in the US? Could we see a compounding interaction between both of them?* These could be part of a "future work recommendation" section of what could be done next from these findings.

Response: We have expanded the Discussion in response to your comment. The revised text discusses the following points:

(1) Indian Ocean warming. Although our study focuses on interannual IOB variability, we note that climate change is projected to cause continued and spatially heterogeneous warming of the tropical Indian Ocean, which may alter IOB variability and strengthen its teleconnections (Cai et al., 2014; Gopika et al., 2025; Rao et al., 2012; Sharma et al., 2023).

(2) Soybean yield projections. We added discussion of projected U.S. soybean yield declines, with losses of 30-40% by the end of the century even under low-emission scenarios (Schlenker and Roberts, 2009) and further reductions under a high-emissions scenario (Hultgren et al., 2025).

(3) Compounding interactions. We highlight the possibility that enhanced IOB variability may interact with a more drought-prone U.S. climate, creating compound risks for soybean production.

(4) Future research. We suggest using coupled climate-crop models to assess these interactions and emphasize adaptation strategies, as well as integrating IOB monitoring into early-warning systems.

3. **Reviewer Comment:** Finally, I would suggest for the code to be made openly available.

**Response:** We will make the code openly available and provide the access link in the Data Availability section of the revised manuscript.

Minor comments:

Line 26: According to FAOSTAT, Brazil has been the main soybean producer for the past years.

**We agree that, according to FAOSTAT, Brazil has been the leading soybean producer in recent years, particularly after 2018. However, our study focuses on the period 1978–2019, during which the United States consistently remained the largest soybean producer until Brazil's recent overtaking. To avoid confusion, we have added a figure (Supplementary Figure S1) comparing U.S. and Brazilian soybean production trends, which highlights that the U.S. was the dominant producer throughout most of our study period, with Brazil surpassing only in the very last years.**

Line 56: there are different verbal tenses on the same paragraph (past and present), I recommend sticking to one for consistency.

**We have revised this paragraph.**

Line 58: a matter of personal taste, but I find adjectives like "valuable" unnecessary in a scientific article.

**We have removed this word.**

Line 59 "food securety"

**We have revised this word.**

Line 84: This is a matter of personal preference, but it's more common to define precipitation as "Pr" or "Precip" than "Pre"

**We have revised the notation and now use "Precip" to represent precipitation throughout the manuscript for consistency.**

Line 128: can you explain explicitly in the text the logical jump (coefficient of determination (R²)) between the -0.41 corr and the 16% variability?

**In the original text, we reported that the correlation coefficient of –0.41 corresponds to 16% of the variability. This comes from the relationship $R^2 = r^2$ in simple linear regression, where $r = -0.41$ gives $R^2 = 0.1681 \approx 0.16$. To be more precise, we have revised the manuscript to report the exact value of 16.8% instead of the rounded 16%.**

Figure 2: Y axis "Values" is not informative enough.

**We have updated the y-axis label in Figure 2 to "Yield Change (% per σ)" to provide a clear and informative description of the plotted values.**

Line 213: Can you improve the clarity of the correlation sentence?

**We have revised this sentence.**

**Reference**

Cai, W., Santoso, A., Wang, G., Weller, E., Wu, L., Ashok, K., Masumoto, Y., Yamagata, T., 2014. Increased frequency of extreme Indian Ocean Dipole events due to greenhouse warming. Nature 510, 254–258. https://doi.org/10.1038/nature13327

Ergo, V.V., Lascano, R., Vega, C.R.C., Parola, R., Carrera, C.S., 2018. Heat and water stressed field-grown soybean: A multivariate study on the relationship between physiological-biochemical traits and yield. Environmental and Experimental Botany 148, 1–11. https://doi.org/10.1016/j.envexpbot.2017.12.023

Fan, J., McConkey, B., Wang, H., Janzen, H., 2016. Root distribution by depth for temperate agricultural crops. Field Crops Research 189, 68–74. https://doi.org/10.1016/j.fcr.2016.02.013

Gaupp, F., Hall, J., Hochrainer-Stigler, S., Dadson, S., 2020. Changing risks of simultaneous global breadbasket failure. Nat. Clim. Chang. 10, 54–57. https://doi.org/10.1038/s41558-019-0600-z

Giraud, L., Langou, J., Rozloznik, M., 2005. The loss of orthogonality in the Gram-Schmidt orthogonalization process. Computers & Mathematics with Applications 50, 1069–1075. https://doi.org/10.1016/j.camwa.2005.08.009

Gobin, A., Van de Vyver, H., 2021. Spatio-temporal variability of dry and wet spells and their influence on crop yields. Agric. For. Meteorol. 308, 108565. https://doi.org/10.1016/j.agrformet.2021.108565

Gopika, S., Sadhvi, K., Vialard, J., Danielli, V., Neetu, S., Lengaigne, M., 2025. Drivers of Future Indian Ocean Warming and Its Spatial Pattern in CMIP Models. Earth's Future 13, e2025EF006112. https://doi.org/10.1029/2025EF006112

Hamed, R., Van Loon, A.F., Aerts, J., Coumou, D., 2021. Impacts of compound hot–dry extremes on US soybean yields. Earth Syst. Dynam. 12, 1371–1391. https://doi.org/10.5194/esd-12-1371-2021

Harris, I., Osborn, T.J., Jones, P., Lister, D., 2020. Version 4 of the CRU TS monthly high-resolution gridded multivariate climate dataset. Sci Data 7, 109. https://doi.org/10.1038/s41597-020-0453-3

Hersbach, H., Bell, B., Berrisford, P., Hirahara, S., Horányi, A., Muñoz-Sabater, J., Nicolas, J., Peubey, C., Radu, R., Schepers, D., Simmons, A., Soci, C., Abdalla, S., Abellan, X., Balsamo, G., Bechtold, P., Biavati, G., Bidlot, J., Bonavita, M., De Chiara, G., Dahlgren, P., Dee, D., Diamantakis, M., Dragani, R., Flemming, J., Forbes, R., Fuentes, M., Geer, A., Haimberger, L., Healy, S., Hogan, R.J., Hólm, E., Janisková, M., Keeley, S., Laloyaux, P., Lopez, P., Lupu, C., Radnoti, G., De Rosnay, P., Rozum, I., Vamborg, F., Villaume, S., Thépaut, J., 2020. The ERA5 global reanalysis. Quart J Royal Meteoro Soc 146, 1999–2049. https://doi.org/10.1002/qj.3803

Hou, Y., Xie, S.-P., Johnson, N.C., Wang, C., Yoo, C., Deng, K., Sun, W., Li, X., 2024. Unveiling the Indian Ocean forcing on winter eastern warming – western cooling pattern over North America. Nat Commun 15, 9654. https://doi.org/10.1038/s41467-024-53921-y

Hultgren, A., Carleton, T., Delgado, M., Gergel, D.R., Greenstone, M., Houser, T., Hsiang, S., Jina, A., Kopp, R.E., Malevich, S.B., McCusker, K.E., Mayer, T., Nath, I., Rising, J., Rode, A., Yuan, J., 2025. Impacts of climate change on global agriculture accounting for adaptation. Nature 642, 644–652. https://doi.org/10.1038/s41586-025-09085-w

Joshi, V.R., Kazula, M.J., Coulter, J.A., Naeve, S.L., Garcia Y Garcia, A., 2021. In-season weather data provide reliable yield estimates of maize and soybean in the US central Corn Belt. Int J Biometeorol 65, 489–502. https://doi.org/10.1007/s00484-020-02039-z

Leng, G., Hall, J., 2019. Crop yield sensitivity of global major agricultural countries to droughts and the projected changes in the future. Science of The Total Environment 654, 811–821. https://doi.org/10.1016/j.scitotenv.2018.10.434

Otkin, J.A., Anderson, M.C., Hain, C., Svoboda, M., Johnson, D., Mueller, R., Tadesse, T., Wardlow, B., Brown, J., 2016. Assessing the evolution of soil moisture and vegetation conditions during the 2012 United States flash drought. Agricultural and Forest Meteorology 218–219, 230–242. https://doi.org/10.1016/j.agrformet.2015.12.065

Rao, S.A., Dhakate, A.R., Saha, S.K., Mahapatra, S., Chaudhari, H.S., Pokhrel, S., Sahu, S.K., 2012. Why is Indian Ocean warming consistently? Climatic Change 110, 709–719. https://doi.org/10.1007/s10584-011-0121-x

Ray, D.K., Gerber, J.S., MacDonald, G.K., West, P.C., 2015. Climate variation explains a third of global crop yield variability. Nat Commun 6, 5989. https://doi.org/10.1038/ncomms6989

Schauberger, B., Archontoulis, S., Arneth, A., Balkovic, J., Ciais, P., Deryng, D., Elliott, J., Folberth, C., Khabarov, N., Mueller, C., Pugh, T.A.M., Rolinski, S., Schaphoff, S., Schmid, E., Wang, X., Schlenker, W., Frieler, K., 2017. Consistent negative response of US crops to high temperatures in observations and crop models. Nat. Commun. 8. https://doi.org/10.1038/ncomms13931

Schlenker, W., Roberts, M.J., 2009. Nonlinear temperature effects indicate severe damages to U.S. crop yields under climate change. Proc. Natl. Acad. Sci. U.S.A. 106, 15594–15598. https://doi.org/10.1073/pnas.0906865106

Sharma, S., Ha, K.-J., Yamaguchi, R., Rodgers, K.B., Timmermann, A., Chung, E.-S., 2023. Future Indian Ocean warming patterns. Nat Commun 14, 1789. https://doi.org/10.1038/s41467-023-37435-7

Zhang, Y., Yang, X., Tian, F., 2024. Study on Soil Moisture Status of Soybean and Corn across the Whole Growth Period Based on UAV Multimodal Remote Sensing. Remote Sensing 16, 3166. https://doi.org/10.3390/rs16173166

---

## Author Response (AR1)

**Response to review of "The Preseason Warming of the Indian Ocean Resulting in Soybean Failure in U.S."**

Dear Editor and Reviewer(s),

We sincerely thank you for your constructive and insightful comments on our manuscript. We have carefully considered all suggestions and revised the manuscript accordingly. Below, we provide a detailed, point-by-point response to each comment. For clarity, reviewer comments are shown in **bold**, our responses are provided in *italics*, and the corresponding revisions in the manuscript are listed under "Manuscript Change".

**Reviewer 1**

**Comment 1: A central assumption of the study is that the IOB-yield relationship reflects a signal that is distinct from ENSO. The manuscript notes that Gram-Schmidt orthogonalization was applied to remove the influence of Niño3.4 from the IOB index, which is a reasonable approach. However, given the apparent Pacific SST anomalies shown in Fig. 6, it would be helpful to clarify the extent to which the IOB signal is statistically independent of ENSO. Was the orthogonalization applied only to the IOB index, or also to the meteorological predictors used in the regression (e.g., Tmax, SMroot)? Could the Pacific anomalies still reflect residual ENSO influence? Since ENSO and Indian Ocean warming often co-evolve, further clarification would be helpful. Aconditional correlation or partial regression analysis (yield vs IOB, controlling for ENSO) would more directly test their statistical independence. If such analyses were not feasible due to sample size or other constraints, a short note acknowledging this would suffice.**

*Response: In our analyses, we applied Gram-Schmidt orthogonalization specifically when examining (i) the correlations between climate indices and U.S. soybean yield, and (ii) the relationships between climate indices, meteorological factors, and atmospheric circulation fields. In both cases, we removed the linear component associated with Niño3.4 from each target variable. In other words, the IOB index, other climate indices, meteorological predictors (e.g., Tmx, SMroot, Pre, VPD), and large-scale circulation variables (e.g., SLP, GPH200, and 925 hPa winds) were all orthogonalized against Niño3.4 before entering the correlation and regression analyses. This ensured that the variability we attributed to the Indian Ocean was linearly independent of ENSO at the same time step.*

*We acknowledge, however, that this approach only guarantees independence at zero lag. Because ENSO and Indian Ocean warming often co-evolve and can exert lead-lag influences on each other across seasons, some residual ENSO-related effects may remain after orthogonalization. We have now added a statement in Section 2.2 to make this limitation explicit. Importantly, the IOB-yield relationship remains significant even after orthogonalization, suggesting that the IOB contributes predictive information beyond ENSO co-variability.*

**Manuscript changes:** In Section 2.2 (Lines 97-99), we revised the text as follows:

B Before conducting the specific analyses, we employed Gram-Schmidt orthogonalization to remove the linear influence of ENSO (represented by Niño3.4) from the IOB index, other climate indices, meteorological factors, and large-scale circulation fields. This method transforms correlated variables into orthogonal sets by sequentially projecting each target variable onto the space orthogonal to ENSO. The ENSO-independent component of a variable $X$ was calculated as:

$$X_{\perp E} = X - \left( \frac{\langle X, E \rangle}{\langle E, E \rangle} \right) E \qquad (5)$$

Where $X$ is the original variable, $E$ the ENSO signal, and $\langle \cdot, \cdot \rangle$ denotes the inner product. Through this procedure, only the variability linearly independent of ENSO is retained, enabling a clearer attribution of Indian Ocean-related effects. We note that while this approach ensures zero-lag statistical independence from ENSO, lead-lag influences cannot be fully eliminated, as ENSO and Indian Ocean warming often co-evolve and interact across seasons. Similar approaches have been applied in recent climate studies (Hou et al., 2024).

**Comment 2: In Fig. 3, Tmax and SMroot emerge as the most influential predictors of soybean yield anomalies. Given that dry soils can lead to elevated Tmax via reduced evaporative cooling, these two variables are often physically and statistically linked. This raises the question of whether they contribute independent information to the regression model or reflect overlapping aspects of the same underlying drought process. Have the authors assessed their correlation or examined variance inflation among predictors? Even a brief note on whether these variables act jointly or additively would help clarify their interpretation within the ridge regression framework.**

*Response: We assessed the correlation between Tmax and SMroot across the U.S. soybean production regions and found a mean Pearson correlation coefficient of -0.42, indicating moderate association rather than strong collinearity. Ridge regression, by design, mitigates multicollinearity through coefficient shrinkage, allowing both predictors to retain meaningful contributions. Tmax primarily captures atmospheric heat stress, whereas SMroot reflects water availability, making them complementary indicators of drought impacts. We have added a clarification in the Discussion to emphasize this interpretation.*

**Manuscript change:** Ridge regression identifies Tmax and SMroot as the most influential meteorological variables for soybean yield anomalies. Tmax primarily reflects atmospheric heat stress, while SMroot measures subsurface water availability and drought persistence. These variables are physically interdependent: dry soils reduce evaporative cooling, increasing Tmax, while higher Tmax enhances evapotranspiration, further reducing SMroot and creating a reinforcing feedback. Despite this interdependence, ridge regression mitigates multicollinearity through coefficient shrinkage, allowing both variables to retain meaningful contributions and jointly characterize compound climate risks. Many other climate variables affecting crop growth also interact, such as soil moisture, precipitation, and temperature, which together determine the water and heat conditions experienced by crops. These complex interactions make yield variability difficult to attribute to any single factor and highlight the importance of modeling approaches that explicitly capture interacting and reinforcing climatic influences on crop production.

**Comment 3: The persistence of soil moisture from winter to summer is a key element of the proposed mechanism. Supplementary Fig. S3 appears to illustrate this, but the discussion could benefit from making more use of it. Would the authors consider highlighting which regions show the strongest ND(-1)J-JAS soil moisture correlation? A short mention in the main text would help readers better understand the spatial aspects of this memory effect.**

*Response: In the original manuscript, we only referenced Supplementary Fig. S3 without explicitly describing the key regions where soil moisture persistence is strongest. In the revised version, we have expanded this section of the Results to highlight that the strongest positive correlations between ND(-1)J (November-December of the year preceding harvest and January of the harvest year) and JAS (July-August-September) soil moisture anomalies are primarily concentrated in the U.S. Midwest and Northern Plains regions that represent the core soybean production areas in the United States. This revision strengthens the connection between early-season hydrological conditions and subsequent summer drought risk, emphasizing the "memory effect" of soil moisture on climate-crop interactions.*

**Manuscript change:** In Section 3.3, we revised the paragraph:

In this study, we found that soil moisture anomalies during ND(-1)J are significantly positively correlated with soil moisture anomalies in JAS (Supplementary Fig. S3). Specifically, the strongest positive correlations are observed over the U.S. Midwest and Northern Plains, which represent the core soybean production regions, indicating that early-season soil moisture conditions in these areas persist into the summer. This "memory effect" suggests that reductions in ND(-1)J soil moisture can influence root-zone moisture during the reproductive stage, thereby affecting soybean yields and highlighting the role of early-season hydrological conditions in modulating summer drought risk.

**Comment 4: The use of a 5-year running mean to detrend soybean yield is a standard choice to remove technological and management-related trends. However, this method may also suppress low-frequency climate variability, such as decadal SST modes, and reduce the number of effective degrees of freedom. Was the sensitivity of the results to this detrending method evaluated? For example, how do key correlations or regression outcomes compare when using a linear detrending approach instead? A brief justification for selecting the 5-year running mean, or a short note on whether this choice meaningfully affects the results, would help readers assess the robustness of the teleconnection signal.**

*Response: We conducted a sensitivity analysis using linear detrending of soybean yield anomalies instead of the 5-year running mean. The correlation with the IOB index changed from -0.41 to -0.389, and the spatial regression results remained largely consistent, albeit with slightly different amplitude. Specifically, the linear detrending yields significant responses in Nebraska, Oklahoma, Mississippi, and Alabama, while the 5-year running means show stronger signals in South Dakota and New York. These differences are modest and do not alter the main conclusions about the IOB's teleconnection pattern. We chose the 5-year running mean as it effectively removes long-term agronomic and technological trends while retaining interannual climate variability; this method is widely used in crop-climate studies (Iizumi et al., 2021)*

*We are also prepared to include the linear detrended regression map as a Supplementary Figure if the reviewer believes it would strengthen the presentation.*

**Comment 5: Pearson correlation is used extensively throughout the manuscript to assess relationships among SST indices, meteorological variables, and anomalies in soybean yield. While this is a standard approach, Pearson correlation assumes linearity and normality, and can be sensitive to outliers. Were these assumptions checked in the analysis? For key relationships such as IOB-yield or SMroot-yield, would the results be consistent if Spearman rank correlation were used instead? Even a brief mention of this in the methods or supplement would help confirm the robustness of the reported associations.**

*Response: To ensure that our conclusions are not biased by the assumptions of Pearson correlation (linearity, normality, and sensitivity to outliers), we repeated the key analyses using Spearman rank correlation, which is non-parametric and more robust to outliers. The results were highly consistent: for instance, the correlation between the ND(-1)J IOB index and U.S. soybean yield anomalies was -0.38 (p=0.017) with Spearman correlation, closely matching the Pearson result (-0.41, p=0.0098). These results confirm that the reported teleconnection is not sensitive to the choice of correlation metric.*

*A supplementary figure summarizing the Spearman results can be provided upon request if the editor deems it useful.*

**Comment 6: The manuscript describes a compelling multi-step pathway: IOB warming leads to changes in atmospheric circulation, reduced soil moisture, increased summer heat and drought, and ultimately, yield loss. Would the authors consider adding a simple schematic to summarize this mechanism? This could help readers from interdisciplinary fields quickly grasp the whole story.**

*Response: We agree that a schematic would be valuable and will add it as a new figure (Figure 7) in the revised manuscript to illustrate the multi-step mechanism linking IOB anomalies to U.S. soybean yield variability.*

**Specific comments**

**L59. "food securety" → "food security".**

*We have revised this word.*

**L66. "Political units" could be ambiguous to international readers. Please specify that this refers to U.S. states.**

*To avoid ambiguity, we revised the wording in the text to explicitly state "U.S. states" instead of "political units."*

**L97. The Gram-Schmidt procedure is mentioned but not described in detail. Clarify whether Niño3.4 was regressed from IOB or vice versa and consider including a short equation or citing a standard reference.**

*We have expanded the Methods (Section 2.2).*

**L126. When selecting ND(-1)J as the optimal window, indicate whether a formal selection criterion (e.g., max correlation, statistical threshold) or multiple testing adjustment was applied.**

*We have clarified our selection procedure. The ND(-1)J period was chosen because it showed the highest absolute Pearson correlation with U.S. soybean yield anomalies among all tested 3-month windows. We did not apply multiple testing corrections because the selection step served as a screening procedure rather than a formal hypothesis test. This clarification has been added to the Results.*

**Manuscript change:** We revised the sentence as:

"The year-to-year anomalies in soybean yield exhibit the strongest Pearson correlation (-0.41) with the IOB index during ND(-1)J (November and December of the year preceding harvest and January of the harvest year), which was identified as the optimal 3-month window following an exhaustive correlation screening across all possible periods; this relationship is statistically significant at the 99% confidence level (two-tailed t-test; Fig. 1)."

**L201-209. Please update figure references to follow the standard format:**

**L201. "Fig. 4(b) and 4(c)" → "Figs. 4(b) and 4(c)"**

**L208. "Fig. 4(b) and 4(e)" → "Figs. 4(b) and 4(e)"**

**L209. "Fig. 4(c)-(e)" → "Figs. 4(c)- (e)"**

**L255. The text refers to "Fig. 6(g)", but the panels go only to (f). This should be corrected.**

*We have revised these figure references.*

**Units and labeling: Colorbars in Figs. 3-6 should include clear units (e.g., "% per σ" or "°C per σ"). Consistent labeling will improve readability.**

*The units of the colorbars are already specified in the figure captions. If the reviewer considers it necessary, we are happy to additionally include the units directly on the colorbars in the revised figures to further improve clarity and consistency.*

**Reviewer 2**

**Comment 1: More details are needed on the Gram-Schmidt orthogonalisation method, it is too**

**vague for now. Among questions and points that I would like to see explained: Can you explain what it does and how you applied it? Can you expand the motivation for this (or what would happen without this step)? Could you be missing signal or information by doing that (special attention to ENSO here)? And if this is a common approach / which other studies have done this before?**

*Response: We have substantially expanded the explanation of Gram–Schmidt orthogonalization in the Statistical analyses (Section 2.2). Specifically, Gram–Schmidt orthogonalization is a standard technique in linear algebra that provides a straightforward framework for transforming a set of potentially correlated variables into an orthogonal (uncorrelated) set by sequentially projecting each variable onto the orthogonal space of the previously processed ones (Giraud et al., 2005). This approach was applied to remove the linear ENSO (Niño 3.4) signal from our climate indices before further analyses. It ensures that subsequent correlation and regression analyses isolate the Indian Ocean effects independently of ENSO.*

*We acknowledge that Gram-Schmidt orthogonalization has potential limitations, such as its dependence on the ordering of variables and sensitivity to numerical instability in the presence of multicollinearity. However, since we only orthogonalized ENSO, the ordering issue is minimized. Nevertheless, we note that the method guarantees independence only at zero lag, so lead-lag interactions between ENSO and Indian Ocean warming may not be fully removed. To establish precedent, we also refer to recent studies that adopted this approach to control for ENSO influences in climate analyses (Hou et al., 2024).*

**Manuscript changes:** In Section 2.2 (Lines 97–99), we revised the text as follows:

B Before conducting the specific analyses, we employed Gram–Schmidt orthogonalization to remove the linear influence of ENSO (represented by Niño3.4) from the IOB index, other climate indices, meteorological factors, and large-scale circulation fields. This method transforms correlated variables into orthogonal sets by sequentially projecting each target variable onto the space orthogonal to ENSO. The ENSO-independent component of a variable $X$ was calculated as:

$$X_{\perp E} = X - \left( \frac{\langle X, E \rangle}{\langle E, E \rangle} \right) E \qquad (5)$$

Where $X$ is the original variable, $E$ the ENSO signal, and $\langle \cdot, \cdot \rangle$ denotes the inner product. Through this procedure, only the variability linearly independent of ENSO is retained, enabling a clearer attribution of Indian Ocean–related effects. We note that while this approach ensures zero-lag statistical independence from ENSO, lead–lag influences cannot be fully eliminated, as ENSO and Indian Ocean warming often co-evolve and interact across seasons. Similar approaches have been applied in recent climate studies (Hou et al., 2024).

**Comment 2: Can you explain and justify the initial choice of meteorological variables? Is this based on previous studies, do similar studies select the same variables? It reads a bit unclear and arbitrary right now.**

*Response: The meteorological variables were selected based on previous agronomic and climate-yield studies that consistently highlight temperature (including its diurnal range, DTR), precipitation, radiation, soil moisture, and humidity as the dominant drivers of soybean yield variability (Gaupp et al., 2020; Hamed et al., 2021; Joshi et al., 2021; Otkin et al., 2016; Ray et al., 2015; Schauberger et al., 2017). In addition, vapor pressure deficit (VPD) has been widely used as a proxy for atmospheric dryness and crop stress (Ergo et al., 2018). To provide transparency, we now include a summary of all variables, their definitions, sources, and supporting references in the Supplementary Material (Table S1).*

*We also note that, in addition to the widely recognized variables, we included cloud cover (Cld) as an exploratory factor. This variable is less commonly studied in soybean yield analyses, but we considered it relevant due to its potential to affect surface energy balance and crop growth. This rationale is now clarified in the revised manuscript and Supplementary Table S1.*

**Manuscript changes:** In Section 2.1 (Lines 81–92), we rewrote the paragraph as:

To assess the impact of meteorological factors on soybean yields in the United States, we selected ten key variables from the Climatic Research Unit (CRU) TS v4.07 dataset and the ERA5 reanalysis (Harris et al., 2020; Hersbach et al., 2020). These include temperature [maximum (Tmx, °C), mean (Tmp, °C), minimum (Tmn, °C), diurnal temperature range (DTR, °C)], precipitation (Pre, mm·d⁻¹), wet day frequency (Wet, days), cloud cover (Cld, %), downward shortwave radiation flux (DSRF, W·m⁻²), root-zone soil moisture (SMroot, m³·m⁻³; Layer 2, 7–28 cm depth), and vapor pressure deficit (VPD, hPa). Eight of these variables were obtained from CRU, which provides monthly mean gridded data at 0.25° × 0.25° resolution, while SMroot was obtained from ERA5 as a proxy for soybean root water uptake. The choice of variables is consistent with previous studies highlighting the role of temperature, precipitation, radiation, soil moisture, and humidity in soybean yields (Gaupp et al., 2020; Gobin and Van de Vyver, 2021; Hamed et al., 2021; Joshi et al., 2021; Leng and Hall, 2019; Ray et al., 2015; Schauberger et al., 2017), with VPD included as an additional dryness indicator (Ergo et al., 2018). VPD was calculated using the following formulas:

$$e_0 = 6.108\exp\left(\frac{17.27 \times \text{Tmp}}{\text{Tmp}+237.3}\right) \qquad (3)$$

$$\text{VPD} = e_0 - e_a \qquad (4)$$

Where Tmp is the monthly average temperature (°C), and $e_a$ is the average actual vapor pressure (hPa), both from the CRU dataset. $e_0$ represents the monthly mean saturated vapor pressure (hPa). A summary of all variables and references, including units, is provided in Table S1 in the Supplementary.

**Comment 3:** t is not clear in the text to me how root zone soil moisture is obtained or calculated. You refer to the ERA5 dataset, but as far as I am aware, this variable is not available on the ERA5 repository.

*Response: In the original manuscript, we referred to "root-zone soil moisture" (SMroot) but did not provide sufficient detail. SMroot was directly obtained from the ERA5 reanalysis dataset as the volumetric soil water content ($m^3 \cdot m^{-3}$). ERA5 provides soil moisture for four layers (0–7 cm, 7–28 cm, 28–100 cm, and 100–289 cm). For this study, we used Layer 2 (7–28 cm depth), which corresponds to the major root water uptake zone for soybean crops. Previous agronomic studies indicate that soybean roots extract most water from the top 30 cm of soil, especially during the reproductive phase, making Layer 2 a reasonable proxy for root-zone soil moisture(Fan et al., 2016; Zhang et al., 2024).*

**Manuscript change:** In Section 2.1, we rewrote the soil moisture description as: "In addition, root zone soil moisture (SMroot, $m^3 \cdot m^{-3}$) was obtained from the ERA5 reanalysis dataset(Hersbach et al., 2020), using Layer 2 (7–28 cm depth) as a proxy for soybean root water uptake."

**Comment 4:** When comparing IOB with meteorological variables, you extract SLP from CRU but geopotential height at 200 hPa, and wind components at 925 hPa from ERA5. ERA5 also has SLP, so is there a reason for this? I would argue that having all variables from the same source would guarantee consistency. If you decide to keep SLP from CRU, it should be shown how similar it behaves between the two sources.

*Response: We thank the reviewer for carefully checking this point. We would like to clarify that in our analysis, SLP was in fact obtained from ERA5, not from CRU. The reference to CRU in the Methods was a writing error. In the revised manuscript, we have corrected this and now state explicitly that all circulation variables (SLP, GPH200, and wind components) were consistently obtained from ERA5 (Section 2.1). We apologize for the oversight and thank the reviewer for helping us improve the clarity of the manuscript.*

**Manuscript change:** We corrected the description of the data source in Section 2.1. The sentence has been revised to: "All variables were obtained from the ERA5 reanalysis dataset (Hersbach et al., 2020)."

**Comment 5:** The last paragraph of the section 2.2 is confusing. On line 104, number (1), you distinguish between meteorological factors and atmospheric circulation patterns? What exactly do you refer to when you mention atmospheric circulation patterns, this has not been introduced before. Would this be the SLP, GPH200 and the wind components? If so, SLP is not an atmospheric circulation variable, and needs to be corrected. If not, then it would need to be better explained or rewritten to improve clarity.

*Response: We agree that our terminology was not sufficiently clear in the original manuscript. In the revised manuscript, we have clarified this terminology. Specifically, we now explicitly define:*

*Meteorological factors as local surface climate variables that directly affect crop growth (e.g., temperature, precipitation, soil moisture, radiation, and humidity).*

*Atmospheric circulation patterns as large-scale circulation fields that characterize regional and hemispheric circulation variability (e.g., sea-level pressure, geopotential height, and winds).*

*Although we acknowledge that sea-level pressure (SLP) is sometimes grouped as a surface variable, in this study we treat SLP as part of the large-scale circulation fields because it reflects broad-scale pressure systems and circulation anomalies. This distinction is now clearly stated in the Methods section to avoid confusion.*

**Manuscript changes:** In Section 2.2, we added a sentence: "For clarity, in this study, we define meteorological factors as local surface climate variables that directly affect crop growth (e.g., Tmx, Pre, SMroot, DSRF, and VPD). In contrast, we define atmospheric circulation patterns as large-scale circulation fields that characterize regional and hemispheric variability, including SLP, GPH200, and 925 hPa winds."

**Comment 6:** The results section combines both actual results and contextualisation aspects that should go into the discussion. And as a consequence, the discussion section is rather small and underdeveloped, looking more like a conclusion than a discussion. Based on that, I would suggest to have the discussion considerably expanded, with the main findings properly contextualised there. For example, the authors find DTR to be important for soybean yield using the ridge regression, which is a statistical approach. I'd like to see potential physical explanations for that (after all, DTR is the difference between two other variables, which could mean many things). Also, have other studies found similar or diverging relations between DTR and soybean yields in the area of study? These aspects should be properly discussed (you could move some of the small contextualisation points from the results to the discussion and expand them there into a coherent text).

*Response: We have extensively revised the Discussion. We expanded it to include:*

(1) *A detailed interpretation of the meteorological predictors Tmax and SMroot, explaining their complementary roles in capturing atmospheric heat stress, soil water availability, and nighttime temperature effects.*

(2) *A discussion of the importance of DTR for soybean yield assessments.*

(3) *A clarification of the methodological framework, describing the use of lead-lag correlation and regression to assess climate-agriculture linkages and its relevance to compound and multi-risk analyses.*

(4) *An expanded discussion of the broader implications under climate change, addressing potential changes in IOB variability, compounding interactions, and future research directions.*

*These additions strengthen the physical and agronomic interpretation of our results.*

**Comment 7:** I also missed the theoretical implications of the findings: what does it mean to have IOB index influencing soybean variability (beyond the practical point of using it to monitor it in advance)? For instance, can it have any interactions with other major climate phenomena, such as climate change? While this is not the focus of the paper, it could still be briefly discussed. Ex: *What are the future projections for the IOB index? What are the future projections for soybean production in the US? Could we see a compounding interaction between both of them?* These could be part of a

**"future work recommendation" section of what could be done next from these findings.**

*Response: We have expanded the Discussion in response to your comment. The revised text discusses the following points:*

> *(1) Indian Ocean warming. Although our study focuses on interannual IOB variability, we note that climate change is projected to cause continued and spatially heterogeneous warming of the tropical Indian Ocean, which may alter IOB variability and strengthen its teleconnections (Cai et al., 2014; Gopika et al., 2025; Rao et al., 2012; Sharma et al., 2023).*

> *(2) Soybean yield projections. We added discussion of projected U.S. soybean yield declines, with losses of 30-40% by the end of the century even under low-emission scenarios (Schlenker and Roberts, 2009) and further reductions under a high-emissions scenario (Hultgren et al., 2025).*

> *(3) Compounding interactions. We highlight the possibility that enhanced IOB variability may interact with a more drought-prone U.S. climate, creating compound risks for soybean production.*

> *(4) Future research. We suggest using coupled climate-crop models to assess these interactions and emphasize adaptation strategies, as well as integrating IOB monitoring into early-warning systems.*

**Comment 8:** Finally, I would suggest for the code to be made openly available.

*Response: We have made the analysis code openly available on GitHub, and the repository link has been added to the Data Availability section of the revised manuscript.*

**Minor comments:**

**Line 26: According to FAOSTAT, Brazil has been the main soybean producer for the past years.**

*We agree that, according to FAOSTAT, Brazil has been the leading soybean producer in recent years, particularly after 2018. However, our study focuses on the period 1978–2019, during which the United States consistently remained the largest soybean producer until Brazil's recent overtaking. To avoid confusion, we have added a figure (Supplementary Figure S1) comparing U.S. and Brazilian soybean production trends, which highlights that the U.S. was the dominant producer throughout most of our study period, with Brazil surpassing only in the very last years.*

**Line 56: there are different verbal tenses on the same paragraph (past and present), I recommend sticking to one for consistency.**

*We have revised this paragraph.*

**Line 58: a matter of personal taste, but I find adjectives like "valuable" unnecessary in a scientific article.**

*We have removed this word.*

**Line 59 "food securety"**

*We have revised this word.*

**Line 84: This is a matter of personal preference, but it's more common to define precipitation as "Pr" or "Precip" than "Pre"**

*We have revised the notation and now use "Precip" to represent precipitation throughout the manuscript for consistency.*

**Line 128: can you explain explicitly in the text the logical jump (coefficient of determination (R²)) between the -0.41 corr and the 16% variability?**

*In the original text, we reported that the correlation coefficient of −0.41 corresponds to 16% of the variability. This comes from the relationship $R^2 = r^2$ in simple linear regression, where $r = −0.41$ gives $R^2 = 0.1681 \approx 0.16$. To be more precise, we have revised the manuscript to report the exact value of 16.8% instead of the rounded 16%.*

**Figure 2: Y axis "Values" is not informative enough.**

*We have updated the y-axis label in Figure 2 to "Yield Change (% per σ)" to provide a clear and informative description of the plotted values.*

**Line 213: Can you improve the clarity of the correlation sentence?**

*We have revised this sentence.*

**Editor**

**Comment: Additionally, please better motivate in your manuscript in what way the applied method is innovative, and how it may be relevant for studies of (other) compound risk and multi-risk. This is important given your submission to the EGU-inter-journal special issue "Methodological innovations for the analysis and management of compound risk and multi-risk, including climate-related and geophysical hazards". Such information can both be included in the introduction section, as well as in the summary/discussion.**

*Response: We have revised the Discussion to clarify the motivation for using the lead-lag correlation and regression framework. This approach was chosen because it can effectively capture both concurrent and delayed relationships between preseason climate variability and crop yields, thereby*

*identifying early predictors of agricultural risk. Lead-lag analysis has been widely applied in climate research to trace causal sequences among large-scale modes of variability, but it has rarely been used to examine climate–agriculture interactions. By applying it here, we demonstrate its usefulness for exploring how multiple climatic drivers act in sequence, which is directly relevant for understanding compound and multi-risk processes. This addition clarifies the rationale for using this framework and its broader relevance to compound-risk analysis.*

**Manuscript changes:** In Discussion, we added a paragraph: "This study employed a lead-lag correlation and regression framework to quantify the influence of preseason climate variability on U.S. soybean yields. This approach captures both concurrent and delayed responses of crop yields to large-scale ocean-atmosphere anomalies, allowing for the identification of early predictors of agricultural risk. Although lead-lag techniques are well established in climate research (Hou et al., 2024; Yang and Xing, 2022; Yang et al., 2025), they have been less frequently applied to quantify climate-agriculture linkages, offering new perspectives on the temporal pathways through which climate variability affects crop production. Furthermore, this framework may be useful for investigating compound and multi-risk events, where multiple climatic drivers interact or occur sequentially. By tracing how early-season anomalies evolve and combine to influence subsequent impacts, it provides a systematic means to explore interconnected climate processes relevant to agricultural and environmental risk assessments."

**Summary of Major Revisions**

1. Methods - Selection of meteorological variables: Rewrote the description of how meteorological factors were selected and added a supplementary table (Table S1) summarizing all variables, including units and references.
2. Methods - Data sources: Clarified the data source of sea level pressure (SLP) as ERA5 and specified the source of soil moisture data.
3. Methods - Statistical analyses: Expanded the description of Gram-Schmidt orthogonalization to provide more methodological detail.
4. Methods - Statistical analyses: Defined meteorological factors and atmospheric circulation patterns explicitly to improve clarity in subsequent analyses.
5. Results (Section 3.1): Revised and expanded the last paragraph of "Climate variabilities and soybean yield anomalies".
6. Results (Section 3.2): Moved the discussion on the interactions between Tmax and SMroot from Results to Discussion for better thematic consistency.
7. Results (Section 3.3): Revised and expanded the section on "Soil Moisture Memory Effect".
8. Discussion: Substantially revised the discussion by:
   a) Adding Figure 7, a schematic illustration of the mechanisms linking IOB warming during ND(-1)J to U.S. soybean yield loss.
   b) Providing a detailed interpretation of the meteorological predictors Tmax and SMroot,

emphasizing their complementary roles in representing atmospheric heat stress, soil water availability, and nighttime temperature effects.

    c) Adding a discussion on the importance of DTR (diurnal temperature range) for soybean yield assessments.

    d) Introducing a paragraph describing the lead-lag correlation and regression framework, highlighting its application to preseason climate-agriculture linkages and its relevance to compound and multi-risk analyses.

    e) Expanding the section on future research directions and implications, including potential changes in IOB variability under climate change and compounding interactions with U.S. drought conditions.

9. Figures: Clarified the y-axis units in all figures to improve readability.

10. Data availability: Added a link to the GitHub repository containing the analysis code.

11. Language editing: Corrected minor grammatical errors and word usage throughout the manuscript.

We hope that these revisions adequately address the reviewers' concerns and improve the clarity and contribution of the manuscript. We appreciate your time and consideration.

Best regards,

Menghan Li, Xichen Li